# Bond Strength and Flexural Capacity of Normal Concrete Beams Strengthened with No-Slump High-Strength, High-Ductility Concrete

**DOI:** 10.3390/ma13194218

**Published:** 2020-09-23

**Authors:** Tian-Feng Yuan, Se-Hee Hong, Hyun-Oh Shin, Young-Soo Yoon

**Affiliations:** 1School of Civil, Environmental and Architectural Engineering, Korea University, 145 Anam-Ro, Seongbuk-Gu 02841, Korea; yuantianfeng@korea.ac.kr (T.-F.Y.); bestshhong@korea.ac.kr (S.-H.H.); 2Department of Agricultural and Rural Engineering, Chungnam National University, 99 Daehak-ro, Yuseong-gu, Daejeon 34134, Korea; hyunoh.shin@cnu.ac.kr

**Keywords:** no-slump, high-strength, high-ductility concrete, normal-strength concrete beams, roughness, interfacial shear strength, flexural strengthening

## Abstract

This study investigates the flexural behavior of normal-strength concrete (NSC) beams that were strengthened with no-slump, high-strength, high-ductility concrete (NSHSDC). A set of slant shear tests was performed to investigate the initial performance of the NSC substrate strengthened with NSHSDC. Slant shear tests considered two types of roughness of interface and five angles of the interface between NSC and NSHSDC. The test results showed that except for specimens with a 75° interface angle, the specimens with high roughness were conformed to the properties (14–21 MPa for 28 days) of the ACI Committee 546 recommendation. For flexural strength tests, NSC beams strengthened with an NSHSDC jacket on the top and bottom sides, three sides, and four sides resulted in strength increments of about 8%, 29%, and 40%, respectively, compared to the beams without NSHSDC jacket. Therefore, the use of NSHSDC is an effective method to improve the performance of NSC beams and is recommended for strengthening reinforced concrete members.

## 1. Introduction

Reinforced concrete is the most widely applied material for construction. Its multifunctionality, economy, and ability to be formed and finished in various shapes make it a very appropriate construction material. However, due to human errors, material disadvantages or changes in environmental criteria, a lot of structures need repairs and strengthening over their design service life. Strengthening of reinforced concrete structures has become very important not only for deteriorated reinforced concrete structures, but achieved much better under service. Previous research studies have used various types of concrete to strengthen reinforced concrete members, such as shotcrete jacketing, laminate jacketing, epoxy bonding, etc. [1,2,3]. However, all these methods have problems with bond strength of concrete to concrete, durability, ductility, and installation problems.

In order to solve those problems, the fiber-reinforced concrete with high bond strength properties was widely developed and used [2,3,4,5,6,7,8,9,10,11]. In recent years, researchers have evaluated the use of ultra-high-performance fiber-reinforced concrete (UHPC) for retrofitting and strengthening of reinforced concrete members [2,3,4,5,6]. Studies published so far have seen promising results in durability and structural performance of UHPC [7,8,9]. Hussein et al. [6] evaluated the shear capacity of UHPC using normal- (NSC) and high-strength (HSC) concrete beams, and the bond strength between these two concrete material layers was significantly high, thus rendering shear connectors unnecessary. Tanarslan [2], Al-Osta et al. [3], Carbonell et al. [4], and Lampropoulos et al. [5] reported that beams strengthened with a different method by UHPC showed an improved yielding and ultimate capacity for all strengthened beams. Habel et al. [10] evaluated the flexural response of full-scale concrete beams reinforced with UHPC, which were cast with UHPC layer in tension. Test results revealed that the UHPC layer significantly improved flexural capacity of the beams. Mohammed et al. [11] reported that NSC beams without any shear reinforcement but just strengthened with a different method using UHPC showed a significant improvement in torsional strength, with a test beam strengthened on four sides showing the highest increase. Whereas, according to their ultra-high strength and low strain capacity, there is an increasing reduction in the overall ductility of retrofitted beams, as their behavior starts to resemble more that of over-reinforced concrete beams. Hence, it is necessary to develop new materials that have outstanding properties of UHPC using retrofitting and strengthening of reinforced concrete members to improve both strength and ductility. However, there are some typical strengths and weaknesses, which not only have high rheological properties and viscosity lead to hardly casting and demolding in a short time, but necessary provide a high-temperature (steam or water) curding condition to achieve strength properties.

A more recent material called no-slump, high-strength, high-ductility concrete (NSHSDC) has been developed [12,13], which has high shape-holding ability without a high-temperature curing condition. The material properties of NSHSDC were investigated to confirm the effectiveness of the mixture design approach and demonstrate the feasibility of combing both high strength (>120 MPa) and ductility (*ε_c_* > 2.9%) into no-slump concrete without heat treatment. The tensile strain capacity and toughness of NSHSDC were 2.8–4.4 times and 0.60–0.76 higher than those of typical UHPC composites [14,15], respectively. NSHSDC showed approximately 76.0% higher tensile strength, and 53.7% lower tensile strain capacity than typical engineering cementitious composites [16].

In this study on using NSHSDC to repair and strengthen NSC beams, it can be noted that none of the research evaluates the individual contribution of longitudinal side strengthening to the flexural strength of beams. NSHSDC can be used for strengthened NSC beams is but lacking in the reference; therefore, the effect of compressive strength of NSC overlay NSHSDC and roughness of interface on slant shear bond strength was evaluated first. Then, the individual (bottom side strengthening) as well as combined (top and bottom sides, two vertical-bottom sides, four sides) effects of jacketing of NSC beams with NSHSDC were evaluated.

## 2. Materials and Methods

### 2.1. Materials

To evaluate the bond strength of normal strength concrete (NSC) and NSHSDC interfaces, the mix proportions were based on 28 days of compressive strength of 45 and 120 MPa, as shown in Table 1 and Table 2. TypeI Portland cement (SSangyong, Seoul, the Republic of Korea) and crushed aggregate (fine and coarse aggregate) were used in the NSC mixture. In the NSHSDC mixture, the TypeI Portland cement (SSangyong, Seoul, the Republic of Korea), silica fume (Elkem Micro silica, Svelgen, Norway), silica filler (SAC Corporation, Ulsan, the Republic of Korea), silica sand with a diameter ranging from 0.08–0.30 mm (SAC Corporation, Ulsan, the Republic of Korea), and 1.5 vol.% of fiber (hybrid using 1.0 vol.% of high-strength steel fiber and 0.5 vol.% of high-strength polyethylene fiber) were used [13]. The chemical and physical properties of these materials are listed in Table 3 and Table 4. The compressive strength tests results, which were evaluated in accordance with ASTM C39 [17] and four-point flexural strength test results, which were evaluated in accordance with ASTM C1609 [18], are shown in Table 5, The tensile strength of NSC was measured by a splitting tensile strength test (ASTM C496 [19]), whereas that of NSHSDC was measured by a direct tensile test, which used dog-bone-shaped specimens based on the JSCE (Japan Society of Civil Engineers, Tokyo, Japan) recommendations [12,13].

### 2.2. Test Method

#### 2.2.1. Interface Bond Strength Characterization Test

The shear strength of the interface was evaluated with slant shear tests, conducted according to ASTM C882 [20], using prismatic specimens of 100 mm × 100 mm × 300 mm, with the interface at four angles (30°, 45°, 60°, and 75°) to the vertical. The interfacial performance of NSC strengthened with NSHSDC was evaluated with a slant shear test on a different angle of the interface, as shown in Table 6.

All specimens were cast on the same day, and the molds were stripped off 24 h after casting. The surface of the concrete substrate was processed to a certain degree of roughness. The repairing surfaces were roughened into two types, low and high, during the first two days of casting. The halves of the specimens were cast by NSC and the surface was cleaned of any extra dust or particles after 24 h, and then NSHSDC was cast to complete the specimens. This was called low roughness category in this study.

There are many methods to achieve high-roughness, such as sandblasting, drilling holes, surface chiseling, steel brushing, and high-pressure water jet spraying [21,22]. However, there are few specialized equipment available for rapid repairs, such as sandblasting and jetting machines. Therefore, the method of surface chiseling (electronic breaker) were adopted in this study since this approach does not require professional operators. The surfaces were completely cleaned, and then NSHSDC was applied to the specimen’s surfaces as shown in Figure 1. All specimens were cured in a room at a temperature of 20 ± 1 °C and humidity of 60 ± 5% until the test day.

#### 2.2.2. Beam Flexural Strength Test

Six reinforcement concrete beams were prepared. The reinforcement concrete beams were 125 mm × 250 mm in cross-section, 2222 mm in length, and reinforced with two D19 bars at the bottom (B-NN). Based on B-NN, the specimen was designed according to the minimum shear requirements of ACI 318–14 and was composed of 8 mm round and smooth steel bars. They were conventional U-shaped open stirrups and spaced at 101 mm along the shear span of the specimen (B-SR).

The other four types beams were distinguished by different strengthening configurations, which are bottom side jacketing (B-BJ), top and bottom sides jacketing (B-2J), three sides jacketing (B-3J), and four sides jacketing (B-4J), respectively. The surface of the NSC concrete beams were chiseled to an average depth of 40 mm on the bottom side and 20 mm on the two vertical sides as well as top side, and then NSHSDC were cast around it inside a mold. All test beams were also cured in a room at a temperature of 20 ± 1 °C and humidity of 60 ± 5% until the test day. The details are shown in Figure 2 and Figure 3. The beams were tested under a three-point loading condition, and the load cell, LVDTs, and steel strain gages were used to the measured applied load, displacement at mid-span, and strain of reinforcement, respectively. The three-point load was applied by a 1000-kN maximum capacity universal testing machine (UTM) (UTM/MTS, 815, Minneapolis, MN, USA) under displacement control at a rate of 0.25 mm/min. Figure 4 shows the details of instrumentation used in this study.

## 3. Results and Discussion

### 3.1. Slant Shear Strength of the Interface

#### 3.1.1. Slant Shear Strength and Failure Mode of Composite NSC and NSHSDC Specimens

Table 7 presents the slant shear test results at 28 days. The failure mode of the slant shear test specimens was divided into interface sliding failure, near interface concrete cracking, and total concrete crushing according to experimental observation [23,24,25]. Compressive strength (*f_ck, s_*) of composite specimens was calculated under maximum load divided by the section area base on the slant shear test, and each of the specimens was fabricated and evaluated the intensity and mean values of the three were shown together. The normal stress acting on the bond interface (*σ*) and shear stress acting on the bond interface (*τ*) was calculated by Equations (1) and (2).
*σ = (Pcos2β)/A*(1)
*τ = (Pcosβsinβ)/A*(2)
where *P* is the ultimate load, *β* is the angle of interface, and *A* is the slant section area.

The nominal horizontal shear strength calculated by ACI 318–19 [26], AASHTO-LRFD [27], CSA [28], and MC2010 [29]. ACI 318–19 is not only ignored friction mechanism but also ignored vertical stress of acting surface to use the same interface shear strength. Whereas, AASHTO-LRFD, CSA, and MC2010 are considered some mechanisms, which are a fraction of concrete strength available to resist interface shear, limiting interface shear resistance, strength reduction factor, concrete safety factor, and design concrete compressive strength, respectively. Therefore, there were occurred lager error range values (***τ/τ_ACI_***), which range was 5.5 to 38.3, as shown in Table 7.

The significant failure mode was shown in Figure 5. According to the test results, total concrete crushing occurred in the specimens with 30° interface angle no matter the roughness of NSC. Near interface concrete cracking occurred in the specimens with interface angle of 45° to 75° at primary station due to their high bond strength between NSC and NSHSDC. Among these, the specimens with the interface angle of 45° exhibited total concrete crushing at the secondary station (final station), whereas the specimens with the interface angle over 60° exhibited interface sliding failure at the secondary station. It is concluded that interface shear strength increased with decreasing interface angle and failure at the total concrete crushing, which is mainly related to concrete failure mode.

Figure 6 shows load versus vertical displacement behavior of the slant shear test specimens. A test was conducted using a universal testing machine with a maximum capacity of 2500 kN, and two LVDTs were installed to measure the vertical displacement. The compressive strength of integral prism specimens (*f_c, NSC_*), constructed entirely with NSC, is also presented in Figure 6 using a dotted border line. All the specimens exhibited linear load–displacement behavior before maximum load, and sudden decreased load related to the failure mode. Interface roughness and angle mainly influence interface strength. It clear that slant shear strength increased with surface roughness, which was approximately 27% than the low roughness specimen. This is attributed to the high roughness increase interaction between NSC and NSHSDC which increases slant shear strength. The specimens with high interface roughness and the lowest interface angle (30HS) exhibited the nearest values to *f_c, NSC_*. This result demonstrated that the interface roughness of concrete is a very effective parameter for determining slant shear strength, which coincides with the results from a previous study [22,30,31].

#### 3.1.2. Comparison of Slant Shear Strength with Design Codes

Figure 7 shows the interfacial bond strength recommended by ACI 546 [32] compared with the test values, which was calculated under maximum load divided by the slant section area base on the slant shear test. Except for the specimens with a 75° interface angle, the specimens with high roughness conformed to the properties (14–21 MPa for 28 days) of the ACI Committee 546 recommendation. For the specimens with no roughness and the interface angle of 30–45° was conformed to the properties of the ACI 546 recommendation. These results verify that high roughening (surface chiseling) is necessary for a strong bond and reliable performance of critical repairs as in the ACI Committee 546 recommendation. These results indicate that the bond between the NSC and NSHSDC surface is great regardless of surface preparation. Furthermore, slant shear strength calculated by current design codes uses a conservative design method, which was totally higher than the test values, as shown in Table 7). This is attributed to difficult prediction of factors affecting slant shear strength, concrete strength, surface handling method, and roughness condition [4,23,33].

### 3.2. Effect of Reinforced Method on Flexural Strength Tests

#### 3.2.1. Load Defection Behavior and Failure Mode

A comparison of the load relative to mid-span deflection behavior for the test beam is shown in Figure 8. Table 8 summarizes the load-deflection properties of tested beams. For B-NN, the first crack occurred as a flexure crack at a load of 11.81 kN (approximately 15.6% of the peak load). The load increased linearly, with a slight reduction in stiffness upon cracking, which occurred shear failure at a load level equal to 75.93 kN (*δ_p_* = 10.17 mm). The B-SR specimen was reinforced by minimum shear stirrups, which was compared with the strengthened beams to estimate the success of the test program, which was also the same with B-NN as the reference beam. It was aimed to observe the flexural strength properties of NSC beams with minimum reinforcement. This specimen initially exhibited the flexural crack at 12.88 kN (approximately 14.1% of the peak load), followed by the first longitudinal reinforcement yielding (*ε_ls_* = 0.024425) at a load of 83.69 kN and a deflection of 10.69 mm. The specimen failed by compressive concrete crushing at the mid-span of the beam when the load and deflection reached to 91.17 kN and 62.40 mm, respectively, as shown in Figure 9.

The B-BJ specimen was strengthened with 40 mm of NSHSDC under the bottom side. This specimen exhibited the first flexural crack at a load of 12.37 kN at the mid-span of the beam and simultaneous new flexural cracks, which mainly distributed at the mid-span of the beam. It is interesting to note that the sound of the debonding of fibers from the NSHSDC matrix could be heard. However, the contribution of NSHSDC (bottom jacketing) to resistance load continued until the sudden shear cracking occurred at 74.04 kN (*δ_p_* = 9.48 mm). Beam B-2J showed a quite similar behavior to the beam B-BJ, exhibiting typical shear failure mode and similar peak load. It is important to note that strengthened NSHSDC did not delaminate from the NSC surface.

In the case of B-3J, the first flexural crack occurred at a load of 13.67 kN (approximately 1.2 times that of B-NN) and the cracks were spread over the 315 mm range of the mid-span of the beam. Among the flexural cracks, a dominant crack developed directly and grew to become the failure crack of the test beam. The contribution of NSHSDC (side and bottom jacketing) to resistance load was continued to increase the load until the compressive concrete crushing occurred at mid-span (the load and deflection reached to 97.71 kN and 15.36 mm) then failed at 41.43 mm of deflection (Figure 10). The B-4J specimen exhibited a large number, distribution, and propagation of cracks when compared to the above-mentioned test beams. The cracks were spread over the 1610 mm range of the mid-span of the beam. This is because of the combined contribution of vertical, top, and bottom sides jacket, as well as increased structure properties, compared to other test beams [3,5,34]. This allowed B-4J to occurred a great performance in peak load, deflection, and crack distribution. It should be noted that NSC strengthens three sides jacketing occurred a considerable loss of ductility [3].

#### 3.2.2. Load–Strain Behavior

The behavior of the load-strain variation in the longitudinal reinforcement was evaluated based on the strains measured by steel strain gages on the rebar surface, as shown in Figure 11. Strain values will be useful to estimate whether a behavior is correct for a given phenomenon, whereby the ultimate behavior of a test beam will be the achievement of the considered stage from the strengthened beam when the strengthening materials correlate with the beam.

The B-NN, B-BJ, and B-2J specimens exhibited shear failure before the yielding of longitudinal reinforcing bars, and these beams showed similar stain values, which were approximately 0.77 × 10^−6^, 0.72 × 10^−6^, and 0.74 × 10^−6^ mm/mm, respectively. This is because NSHSDC did not been operate at full capacity even after the NSC beam exhibited shear cracks at lower load levels. For B-3J, compressive concrete crushing occurred at mid-span and a dominant flexural crack developed directly and grew to become a failure crack of the test beam, as shown in Figure 8 and Figure 9. This is similar behavior compared with B-SR, but the strain values of the B-3J beam, 11.93 × 10^−6^ mm/mm were significantly lower than the yield strain values of the longitudinal reinforcement. It can be noted that the NSC encased in NSHSDC exhibited compressive concrete crushing at 97.35 kN without longitudinal steel yielded. It explains why NSC strengthened by three sides jacketing showed a considerable loss of ductility and a few cracks distributed in the mid-span of test beam (Figure 9). However, B-4J exhibited a similar maximum strain values to B-SR, which was approximately 0.04771 mm/mm at 100.9 kN. Therefore, the top surface of the NSC beam strengthened by NSHSDC can have significantly improved structural properties, which was verified in previous research [2].

#### 3.2.3. Effect of Strengthening Method

This study aimed to evaluate the strengthening contribution of NSHSDC to the flexural behavior of NSC beams. The ultimate strength and behavior were mainly influenced by not only strengthening types but also strengthening methods [2,3]. By considering these parameters, the test plan was fulfilled step by step, which is aimed to improve strength, ductility, and stiffness by strengthening NSHSDC. The results in Figure 12 show that NSC beams strengthened with an NSHSDC jacket on the top and bottom sides (B-2J), three sides (B-3J), and four sides (B-4J) resulted in slight increments in maximum capacity of about 8%, 29%, and 40%, respectively, compared to B-NN. The beams B-3J and B-4J showed approximately 6.7% and 15.9%, respectively—higher maximum loads than B-SR. However, there was a significant problem with the ductility of the test beam, with the failure mode for beams strengthened with the NSHSDC jacket on three sides being an undesirable softening mode post-peak load. There were optimistic results for NSC beams strengthened on four sides with not only improvement in ultimate capacity, but also an increase in energy absorption capacity. Furthermore, the test beam strengthened on three sides showed slightly increase in stiffness. This is because increasing number of strengthening sides (3J and 4J) gets nearer to over-reinforced concrete beams [2]. Furthermore, according to the NSC beam exhibited shear crack in lower load levels leads to NSHSDC has not been operating at full capacity, the B-BJ occurred similar behavior with B-NN.

## 4. Conclusions

An experimental investigation was performed to evaluate the slant shear bond strength and flexural performance of NSC strengthened with NSHSDC. To evaluate the flexural performance, test beams having four different strengthening configurations were constructed and tested under monotonically increasing load. Based on the results of this investigation, the following concluding remarks can be made:(1)Regarding slant shear bond properties, specimens with high roughness conformed to the properties (14–21 MPa for 28 days) of the ACI Committee 546 recommendation, except specimens with 75° of interface angle. For the low roughness specimens, an interface angle of 30–45° conformed to the properties of ACI recommendation. The high roughness specimens exhibited higher interface stresses than those of the low (no) roughness specimens, and the interface shear stress ratios of the test to prediction (τ/τpred) ranged from 1.27 to 2.17.(2)In the effect of the strengthening method in the flexural loading test, beams B-BJ and B-2J exhibited similar behavior as B-NN, which exhibited typical shear failure mode without significantly improve the peak load. However, it is important to note that strengthened NSHSDC did not delaminate from the NSC surface.(3)Specimens B-3J and B-4J exhibited similar behavior to B-SR, except that the ductility of B-3J decreased rapidly with more stress up to the failure of the beam structure. B-4J occurred flexure failure similar to minimum shear reinforcement concrete (B-SR). Additionally, the test beam strengthened from three sides slightly increased in stiffness.

## Figures and Tables

**Figure 1 materials-13-04218-f001:**
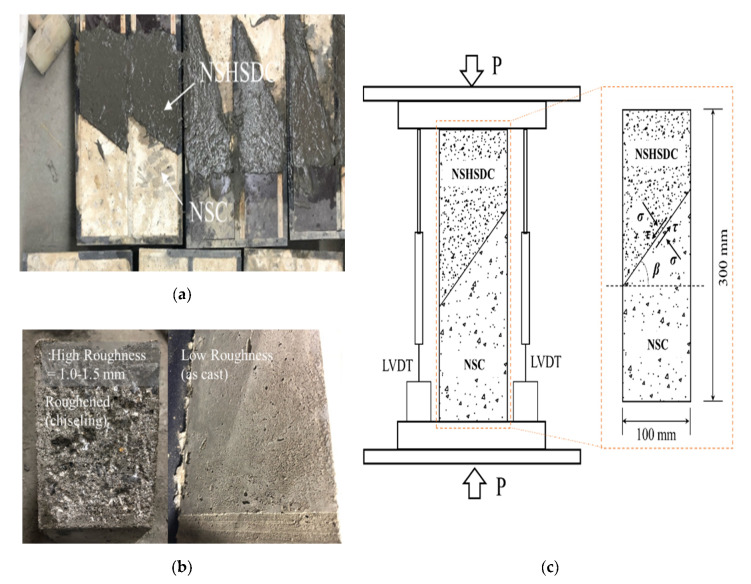
Slant shear test: (**a**) Production of the prismatic specimens, (**b**) Interface condition, (**c**) Test set-up.

**Figure 2 materials-13-04218-f002:**
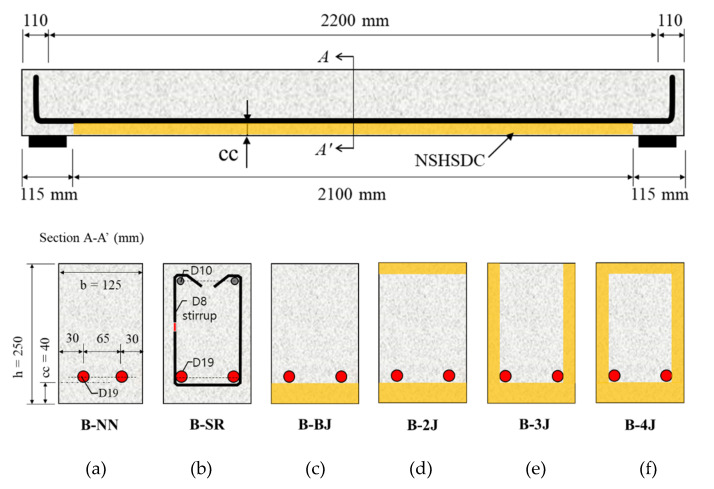
Strengthening schemes of test beams: (**a**) B-NN; (**b**) B-SR; (**c**) B-BJ); (**d**) B-2J; (**e**) B-3J; (**f**): B-4J.

**Figure 3 materials-13-04218-f003:**
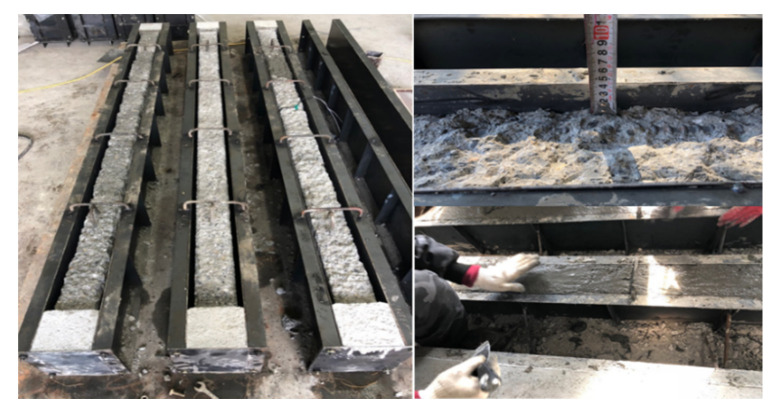
Strengthening process using chiseling and NSHSDC cast in situ strengthening technique.

**Figure 4 materials-13-04218-f004:**
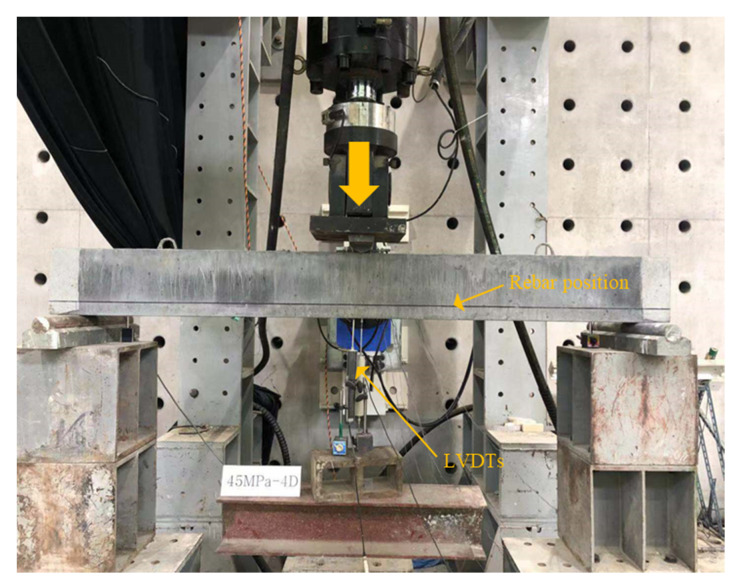
Test set-up of beam flexural strength test.

**Figure 5 materials-13-04218-f005:**
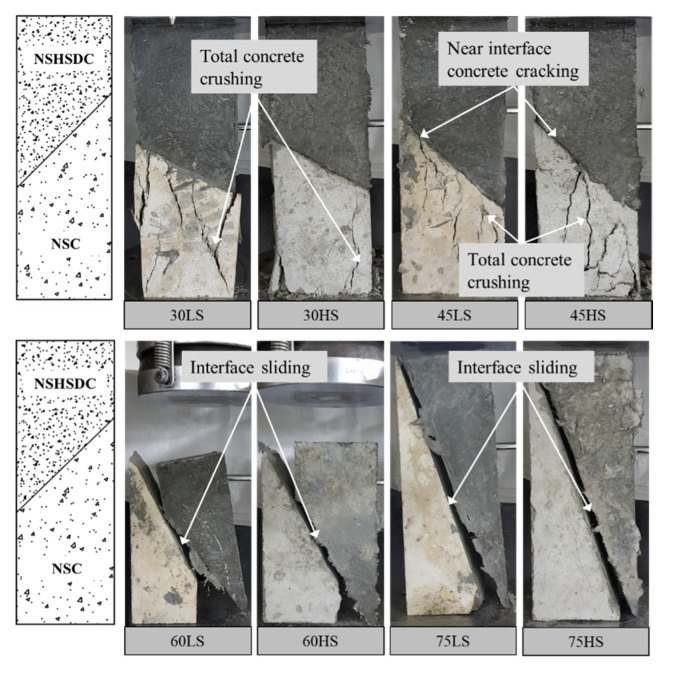
Failure mode of slant shear strength test.

**Figure 6 materials-13-04218-f006:**
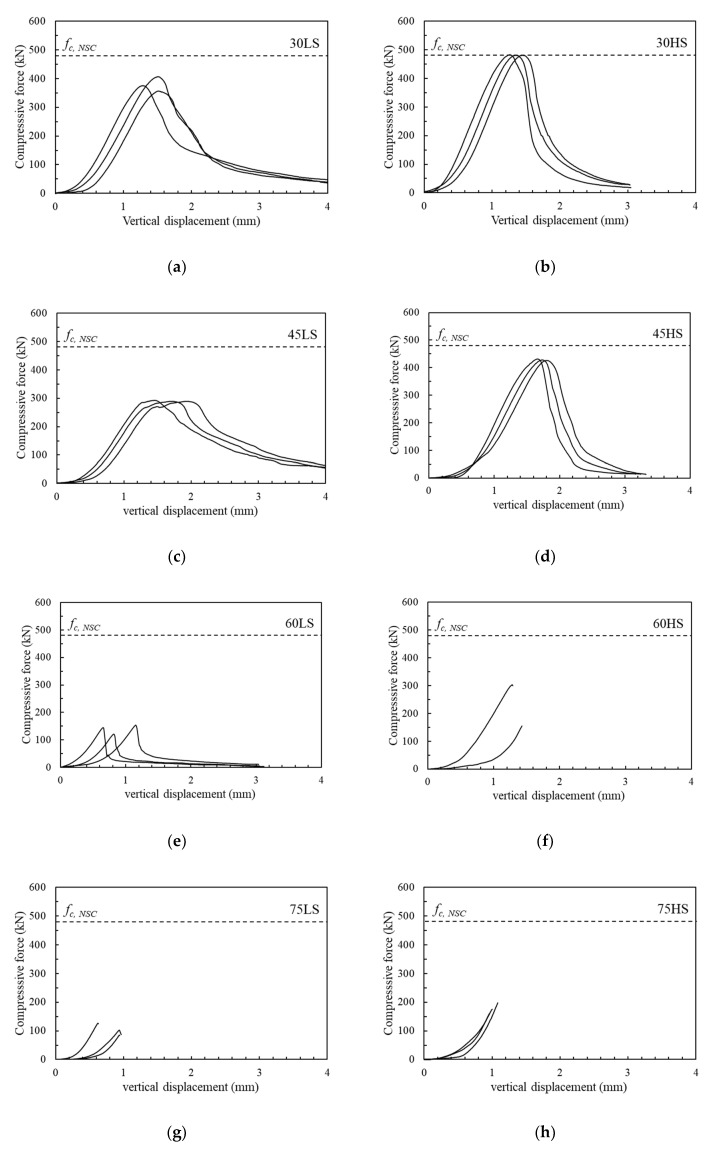
Load–vertical displacement behavior of slant shear strength test: (**a**) 30LS; (**b**) 30HS; (**c**) 45LS; (**d**) 45HS; (**e**) 60LS; (**f**) 60HS; (**g**) 75LS; (**h**) 75HS.

**Figure 7 materials-13-04218-f007:**
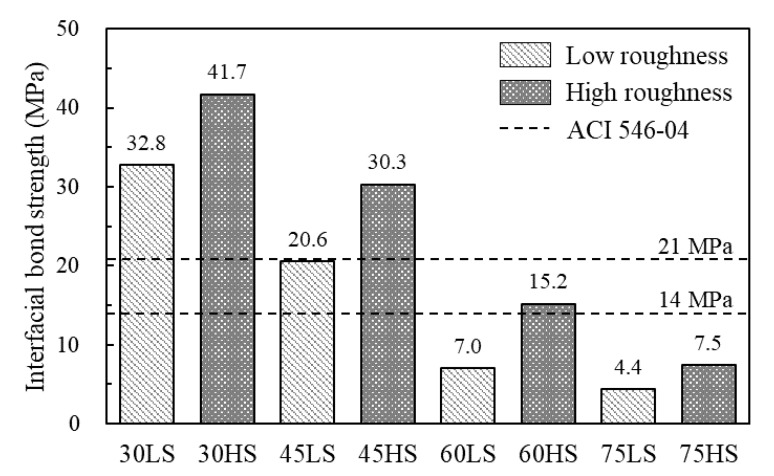
Interfacial bond strength behavior.

**Figure 8 materials-13-04218-f008:**
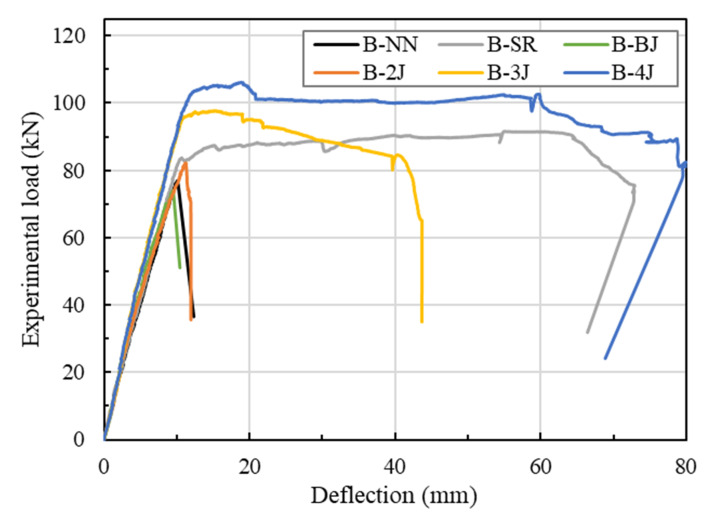
Load–deflection curves of test beams.

**Figure 9 materials-13-04218-f009:**
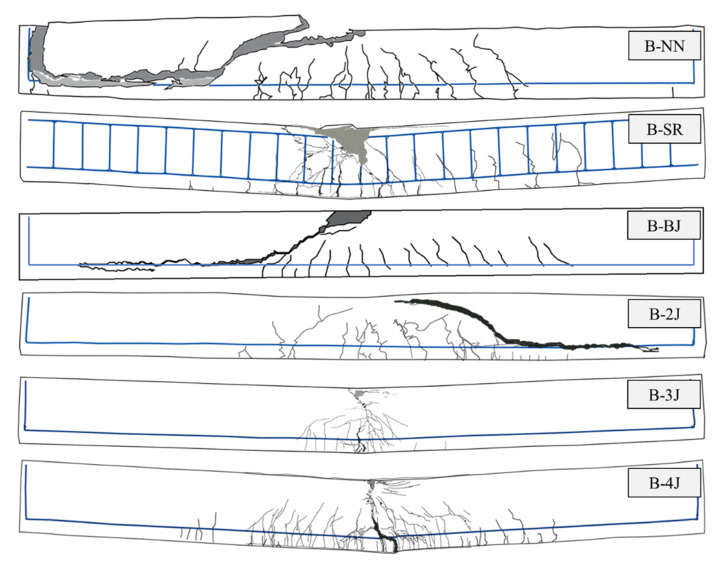
Crack distribution and failure mode of test beams.

**Figure 10 materials-13-04218-f010:**
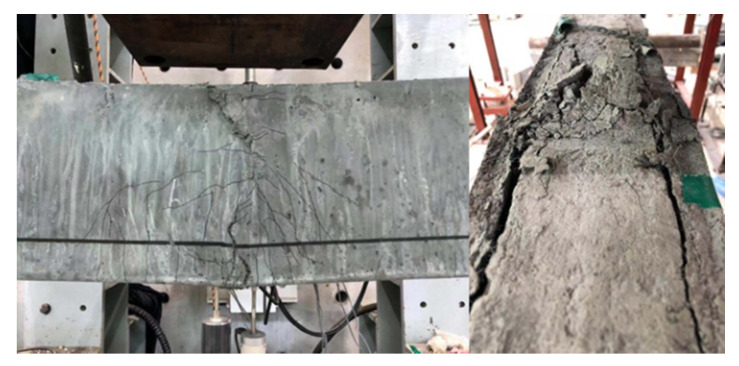
Compressive failure (B-3J).

**Figure 11 materials-13-04218-f011:**
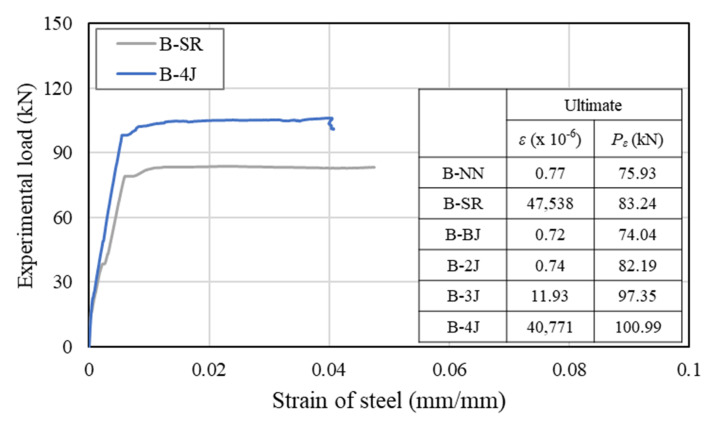
Load–strain of longitudinal steel.

**Figure 12 materials-13-04218-f012:**
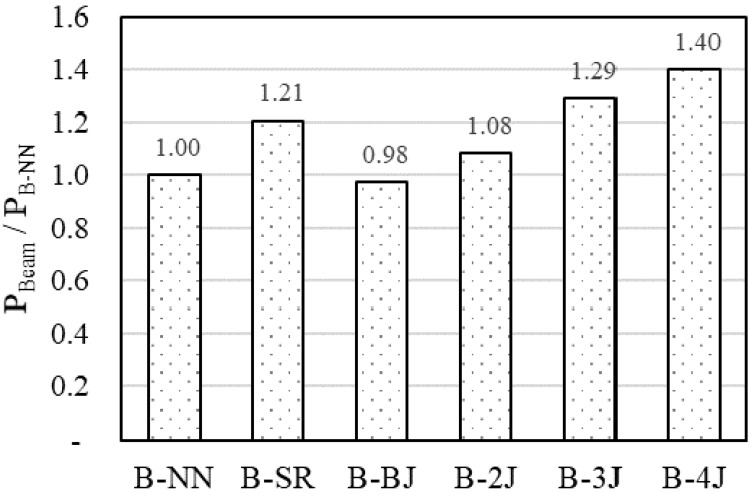
Effect of strengthening method on peak load of test beam.

**Table 1 materials-13-04218-t001:** Mixture proportion of NSC.

Type	*w/c*	Water	Cement	Fine Aggregate	Coarse Aggregate	SP	Slump
NSC	0.43	165	380	818	921	0.8%	175 mm

[Note] NSC = normal strength concrete; *w*/*c* = water to cement ratio; SP = superplasticizer.

**Table 2 materials-13-04218-t002:** Mixture proportion of NSHSDC.

Type	*w/b*	Water	Cement	Silica Fume	Silica Filler	Silica Sand	Steel Fiber	Polyethylene Fiber	SP
NSHSDC	0.172	0.215	1.00	0.25	0.30	1.10	1.0%	0.5%	3.0%

[Note] NSHSDC = no-slump high-strength, high-ductility concrete; *w/b* = water to binder ratio; SP = superplasticizer.

**Table 3 materials-13-04218-t003:** Chemical compositions and physical properties of materials.

Type	Surface Area (cm^2^/g)	Density (g/cm^3^)	Chemical Composition (%)						
			SiO_2_	Al_2_O_3_	Fe_2_O_3_	CaO	MgO	SO_3_	Na_2_O
Cement	3492	3.15	21.2	4.7	3.1	62.8	2.8	2.1	-
Silica fume	200,000	2.20	96.0	0.3	0.1	0.4	0.1	<0.2	-
Silica filler	2.65	0.75	99.6	0.3	0.03	0.01	0.006	-	0.009

**Table 4 materials-13-04218-t004:** Properties of fibers.

Type	Diameter *d_f_*, (mm)	Length *l_f_*, (mm)	Aspect Ratio (*l_f_*/*d_f_*)	Density (g/cm^3^)	Tensile Strength (MPa)	Elastic Modulus (GPa)
High strength straight steel fiber	0.2	19.5	97.5	7.8	2650	200
High strength polyethylene fiber	31 μm	12	387	0.97	2900	100
Hooked-end steel fiber	0.55	35	65	7.9	1400	200

**Table 5 materials-13-04218-t005:** Strength test results (28 day).

Type	Compressive Strength (MPa/CV)	Flexural Strength (MPa/CV)	Tensile Strength (MPa/CV)	Remarks
NSC	46.2/0.1	26.5/0.6	3.9/0.5	Splitting tensile strength test
NSHDC	123.4/0.3	21.9/8.5	9.7/1.6	Direct tensile strength

[Note] CV = coefficient of variation.

**Table 6 materials-13-04218-t006:** Summary of slant shear test specimens.

Specimen	Test Type	Roughness	Angle of Interface (*β*)	Remarks
NSC	Compression	-	-	-
30LS	Slant shear	As cast	30°	Low roughness
30HS	Slant shear	Chiseling	30°	High roughness
45LS	Slant shear	As cast	45°	Low roughness
45HS	Slant shear	Chiseling	45°	High roughness
60LS	Slant shear	As cast	60°	Low roughness
60HS	Slant shear	Chiseling	60°	High roughness
75LS	Slant shear	As cast	75°	Low roughness
75HS	Slant shear	Chiseling	75°	High roughness

**Table 7 materials-13-04218-t007:** Summary of slant shear test results.

Specimen	Failure Mode		*f_ck_, _s_* (MPa)			*σ* (MPa)	*τ* (MPa)	Shear Strength Ratio, *τ/τ_pred_*			
	Primary	Secondary	Mean	Norm	CV			*τ/τ_ACI_*	*τ/τ_ASH_*	*τ/τ_CSA_*	*τ/τ_fib_*
**NSC**	**CC**	-	48.6	1	0.05	-	-	-	-	-	-
30LS	CC	-	37.9	0.78	0.43	28.4	16.4	29.3	0.6	1.5	0.8
30HS	CC	-	48.2	0.99	0.10	36.2	20.9	37.3	0.9	1.6	0.9
45LS	NI	CC	29.1	0.60	0.24	14.6	14.5	26.0	0.9	2.5	1.3
45HS	NI	CC	42.9	0.88	0.49	21.5	21.4	38.3	1.6	2.7	1.6
60LS	NI	IS	14.0	0.29	0.78	3.5	6.1	10.8	1.4	4.0	2.1
60HS	NI	IS	30.4	0.63	0.61	7.6	13.2	23.5	2.3	4.5	2.3
75LS	NI	IS	10.4	0.21	1.62	1.0	3.1	5.5	1.6	5.5	2.3
75HS	NI	IS	17.7	0.36	0.58	1.7	5.3	9.4	2.7	6.5	2.7
Mean								22.5	1.5	3.6	1.7
COV								0.53	0.46	0.48	0.37

[Note] *f_ck, s_* = compressive strength obtained from the slant shear test; CC = total concrete crushing; IS = interface sliding failure; NI = near interface concrete cracking; Mean = mean value; Norm = normalized value by *f_ck, s_* of monolithic specimens (NSC); CV = coefficient of variation; *τ_ACI_*, *τ_ASH_*, *τ_CSA_*, *τ_fib_*, = nominal horizontal shear strength calculated by ACI 318–19, AASHTO-LRFD, CSA, and MC 2010, respectively.

**Table 8 materials-13-04218-t008:** Test results of beams.

Specimen	Initial Crack		Ultimate		Failure		Initial Stiffness (kN·mm)	Deflection Ductility (*δ_p_*/*δ_cr_*)	Energy Absorption (kN·m)
	*P_cr_* (kN)	*δ_cr_* (mm)	*P_p_* (kN)	*δ_p_* (mm)	*P_f_* (kN)	*δ_f_* (mm)			
B-NN	11.81	1.30	75.93	10.17	75.93	10.17	9.08	7.82	0.39
B-SR	12.88	1.43	91.58	55.08	91.17	62.40	9.01	38.82	5.73
B-BJ	12.37	1.43	74.04	9.48	74.04	9.48	8.65	6.63	0.35
B-2J	13.41	1.57	82.19	11.21	82.19	11.21	8.54	7.14	0.46
B-3J	13.67	1.55	97.71	15.36	82.44	41.43	8.82	9.91	3.32
B-4J	13.70	1.51	106.12	18.80	89.21	79.24	9.07	12.45	7.11

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
