# Peer review of "Bond Strength and Flexural Capacity of Normal Concrete Beams Strengthened with No-Slump High-Strength, High-Ductility Concrete"

_materials, 2020, doi:10.3390/ma13194218_

Round 1
Reviewer 1 Report
In this study, the authors used slant shear test and flexural strength rest to investigate interface bond strength and flexural behavior of the NSC substrate strengthened with NSHSDC, respectively. They discussed the effect of different roughness and angles of the interface between NSC and NSHSDC on bond strength, and compared the flexural strength of NSC beams strengthened with NSHSDC jacket on several sides. I suggest the authors may revise the manuscript as follows before the publishing
- As the author mentioned in Test Method, the picture of prismatic specimens with a 0° interface angle should be added in the Figure 5.
- The author should replace Figure 6 to make them have better resolution
- The author should further compare the date in table 7 rather than just a simply summary.
- Some grammar and spelling mistakes should be corrected, and the author should check the article carefully, such as: “A a set of ” in line 18, “ wirh ” in line 211 and so on.
Author Response
In this study, the authors used slant shear test and flexural strength rest to investigate interface bond strength and flexural behavior of the NSC substrate strengthened with NSHSDC, respectively. They discussed the effect of different roughness and angles of the interface between NSC and NSHSDC on bond strength, and compared the flexural strength of NSC beams strengthened with NSHSDC jacket on several sides. I suggest the authors may revise the manuscript as follows before the publishing
Answer: First of all, thank you very much for your useful comments on our paper. We have carefully considered all your comments, and the revised manuscript is now attached for your reconsideration. We really appreciate the opportunity to resubmit. Also, we would like to thank you for your excellent comments which significantly improved the quality of our paper.
1) As the author mentioned in Test Method, the picture of prismatic specimens with a 0° interface angle should be added in the Figure 5.
Answer: Thank you for your recommendation.
The prismatic specimens with 30°, 45°, 60°, 75°, respectively, interface angles were considered in this research. The prismatic specimen made in NSC or NSHSDC was prepared, which just using as compare variables. If necessary I will add this picture in the paper.
2) The author should replace Figure 6 to make them have better resolution.
Answer: As you recommended, the figure is modified.
3) The author should further compare the date in table 7 rather than just a simply summary.
Answer: As you recommended, the information is modified
ACI 318-19 is not only ignored friction mechanism but also ignored vertical stress of acting surface to use the same interface shear strength. Whereas, AASHTO-LRFD, CSA, and MC2010 are considered some mechanisms, which are a fraction of concrete strength available to resist interface shear, limiting interface shear resistance, strength reduction factor, concrete safety factor, and design concrete compressive strength, respectively. Therefore, there were occurred lager error range values (τ/τACI), which range was 5.5 to 38.3, as shown in Table 7.
Furthermore, the details of analysis results shown in figure 7 were mainly explained in figure 5 to 7.
4) Some grammar and spelling mistakes should be corrected, and the author should check the article carefully, such as: “A a set of ” in line 18, “ wirh ” in line 211 and so on.
Answer: As you recommended, the information is modified
Reviewer 2 Report
The authors present an interesting paper on bond strength and flexural capacity of normal concrete beams strengthened with no-slump high-strength high ductility concrete. In general, the paper is well organized and structured, presenting interesting and relevant results. In my opinion, the paper has conditions to be considered for publication after some essential corrections to improve its overall quality.
In the initial part of the paper (in the introduction) the authors should develop in more detail the part referring to the current state of knowledge. In this way, it would be possible to move towards justifying the need for this work in a more evident and secure way.
Tables 1 and 2 should be presented more carefully, including for example the mix proportions in kg/m3.
Chapter 3 in general is very interesting, however it would be appropriate for the results obtained to be compared with values of other authors or even with reference values. Benchmarking is essential in this type of work.
Author Response
The authors present an interesting paper on bond strength and flexural capacity of normal concrete beams strengthened with no-slump high-strength high ductility concrete. In general, the paper is well organized and structured, presenting interesting and relevant results. In my opinion, the paper has conditions to be considered for publication after some essential corrections to improve its overall quality.
Answer: First of all, thank you very much for your useful comments on our paper. We have carefully considered all your comments, and the revised manuscript is now attached for your reconsideration. We really appreciate the opportunity to resubmit. Also, we would like to thank you for your excellent comments which significantly improved the quality of our paper.
1) In the initial part of the paper (in the introduction) the authors should develop in more detail the part referring to the current state of knowledge. In this way, it would be possible to move towards justifying the need for this work in a more evident and secure way.
Answer: As you recommended, the introduction are modified.
In order to solve those problems, the fiber reinforced concrete with high bond strength properties was widely developed and used [2-11]. In recent years, researchers have evaluated the use of ultra-high-performance fiber-reinforced concrete (UHPC) for retrofitting and strengthening of reinforced concrete members [2-6]. Studies published so far have seen promising results in durability and structural performance of UHPC [7-9]. Hussein et al. [6] evaluated the shear capacity of UHPC using normal-strength (NSC) and high-strength (HSC) concrete beams, and the bond strength between these two concrete material layers was significantly high, thus shear connectors unnecessary. Tanarslan [2], Al-Osta et al. [3], Carbonell et al. [4], and Lampropoulos et al. [5] reported that beams strengthened with a different method by UHPC showed an improved yielding and ultimate capacity for all strengthened beams. Habel et al. [10] evaluated the flexural response of full-scale concrete beams reinforced with UHPC, which were cast with UHPC layer in tension. Test results revealed that the UHPC layer significantly improved flexural capacity of the beams. Mohammed et al. [11] reported that NSC beams without any shear reinforcement but just strengthened with a different method using UHPC showed a significant improvement in torsional strength, with a test beam strengthened on four sides showing the highest increase. Whereas, according to ultra-high strength and low strain capacity, there is an increasing reduction in the overall ductility of retrofitted beams, as their behavior starts to resemble more that of over-reinforced concrete beams. Hence, it is necessary to develop new materials that have outstanding properties of UHPC using retrofitting and strengthening of reinforced concrete members to improve both strength and ductility. However, there are some typical strengths and weaknesses, which not only have high rheological properties and viscosity lead to hardly casting and demoulding in a short time, but necessary provide high temperature (steam or water) curding condition to achieve strength properties.
A more recent material called no-slump high-strength, high-ductility concrete (NSHSDC) was developed [12, 13], which have the high shape-holding ability and without high temperature curing condition.
2) Tables 1 and 2 should be presented more carefully, including for example the mix proportions in kg/m3.
Answer: As you recommended, the information is modified.
Unfortunately, the information of NSC is modified as your recommendation, but the information of NSHSDC cannot detail present in this moment according to confidentiality agreements of project. Please understand and forgive.
3) Chapter 3 in general is very interesting, however it would be appropriate for the results obtained to be compared with values of other authors or even with reference values. Benchmarking is essential in this type of work.
Answer: Thank you for your recommendation. There are many part of comparison as your recommendation.
Such as,
The specimens with high interface roughness and the lowest interface angle (30HS) exhibited the nearest values to fc, NSC. This result demonstrated that the interface roughness of concrete is a very effective parameter for determining slant shear strength, which is coincide with the results from a previous study [22, 30, 31].
These results indicate that the bond between the NSC and NSHSDC surface is great regardless of surface preparation. Furthermore, slant shear strength calculated by current design codes uses a conservative design method, which was totally higher than the test values, as shown in Table 7). This is attributed to difficult prediction of factors affecting slant shear strength, concrete strength, surface handling method, and roughness condition [4, 23, 33].
The B-4J specimen exhibited a large number, distribution, and propagation of cracks when compared to the above-mentioned test beams. The cracks were spread over the 1,610 mm range of the mid-span of the beam. This is because of the combined contribution of vertical, top, and bottom sides jacket, as well as increased structure properties, compared to other test beams [3, 5, 34]. This allowed B-4J to occurred a great performance in peak load, deflection, and crack distribution. It should be noted that NSC strengthens three sides jacketing occurred a considerable loss of ductility [3].
It explains the reasons why NSC strengthen three sides jacketing showed considerable loss of ductility and a few numbers of cracks distributed in the mid-span of test beam (Figure 9). However, B-4J exhibited a similar maximum strain values to B-SR, which was approximately 0.04771 mm/mm at 100.9 kN. Therefore, the top surface of the NSC beam strengthened by NSHSDC can have significantly improved structural properties, which was verified in previous research [2].
This study aimed to evaluate the strengthening contribution of NSHSDC to the flexural behavior of NSC beams. The ultimate strength and behavior were mainly influenced by not only strengthening types but also strengthening methods [2, 3].
There were optimistic results for NSC beams strengthened on four sides with not only improvement in ultimate capacity, but also an increase in energy absorption capacity. Furthermore, the test beam strengthened on three sides showed slightly increase in stiffness. This is because increasing number of strengthening sides (3J and 4J) gets nearer to over-reinforced concrete beams [2]. Furthermore, according to the NSC beam exhibited shear crack in lower load levels leads to NSHSDC has not been operating at full capacity, the B-BJ occurred similar behavior with B-NN.
This is first research using NSHSDC reinforced concrete beams, thus, it is just referred to reference mechanism, and hardly compared with reference values. Please understand and forgive.
Round 2
Reviewer 1 Report
The authors have replied all my questions. The paper is ready for publishing.